# Distinct Dopamine D_2_ Receptor Antagonists Differentially Impact D_2_ Receptor Oligomerization

**DOI:** 10.3390/ijms20071686

**Published:** 2019-04-04

**Authors:** Elise Wouters, Adrián Ricarte Marín, James Andrew Rupert Dalton, Jesús Giraldo, Christophe Stove

**Affiliations:** 1Laboratory of Toxicology, Department of Bioanalysis, Faculty of Pharmaceutical Sciences, Ghent University, Ottergemsesteenweg 460, 9000 Ghent, Belgium; elise.wouters@ugent.be; 2Laboratory of Molecular Neuropharmacology and Bioinformatics, Unitat de Bioestadística, Institut de Neurociències, Universitat Autònoma de Barcelona, 08193 Bellaterra, Spain; adrian.ricarte@e-campus.uab.cat (A.R.M.); James.Dalton@uab.cat (J.A.R.D.); Jesus.Giraldo@uab.es (J.G.); 3Instituto de Salud Carlos III, Centro de Investigación Biomédica en Red de Salud Mental, CIBERSAM, Universitat Autònoma de Barcelona, 08193 Bellaterra, Spain; 4Unitat de Neurociència Traslacional, Parc Taulí Hospital Universitari, Institut d’Investigació i Innovació Parc Taulí (I3PT), Institut de Neurociències, Universitat Autònoma de Barcelona, 08193 Bellaterra, Spain

**Keywords:** G protein-coupled receptor (GPCR), dimerization, oligomerization, protein complementation assay, NanoLuc binary technology (NanoBiT), dopamine D_2_ receptor

## Abstract

Dopamine D_2_ receptors (D_2_R) are known to form transient homodimer complexes, of which the increased formation has already been associated with development of schizophrenia. Pharmacological targeting and modulation of the equilibrium of these receptor homodimers might lead to a better understanding of the critical role played by these complexes in physiological and pathological conditions. Whereas agonist addition has shown to prolong the D_2_R dimer lifetime and increase the level of dimer formation, the possible influence of D_2_R antagonists on dimerization has remained rather unexplored. Here, using a live-cell reporter assay based on the functional complementation of a split Nanoluciferase, a panel of six D_2_R antagonists were screened for their ability to modulate the level of D_2L_R dimer formation. Incubation with the D_2_R antagonist spiperone decreased the level of D_2L_R dimer formation significantly by 40–60% in real-time and after long-term (≥16 h) incubations. The fact that dimer formation of the well-studied A_2a_–D_2L_R dimer was not altered following incubation with spiperone supports the specificity of this observation. Other D_2_R antagonists, such as clozapine, risperidone, and droperidol did not significantly evoke this dissociation event. Furthermore, molecular modeling reveals that spiperone presents specific Tyr199^5.48^ and Phe390^6.52^ conformations, compared to clozapine, which may determine D_2_R homodimerization.

## 1. Introduction

Dopamine receptors belong to the class A sub-family of G protein-coupled receptors (GPCRs). Five dopamine receptors have been identified in mammals and are classified in the D_1_-like family, with the D_1_ and D_5_ subtypes, and the D_2_-like family, with the D_2_, D_3_, and D_4_ subtypes [1]. They are key players in the coordination of motor control, cognitive function, memory, and reward [2,3]. A growing body of evidence indicates that the signaling function of many GPCRs is diversified and fine-tuned by interaction with other GPCRs [4]. Dimerization of GPCRs has been demonstrated both in vitro and in vivo, whereby association may take place with the same GPCR (homo-oligomerization) or with different GPCRs (hetero-oligomerization) [5]. Dimerization phenomena have been documented for all five dopamine receptor subtypes [6,7,8,9]. Towards this end, extensive work has been directed towards the dopamine D_2_ sub-type receptor (D_2_R). This receptor plays an important role in the physiological actions of the neurotransmitter dopamine, and it is a target for drugs used to treat schizophrenia and Parkinson’s disease, depression, attention deficit hyperactivity, stress, nausea, and vomiting [10,11,12,13,14,15,16]. 

The D_2_R exists in two isoforms, D_2,short_ (D_2S_R) and D_2,long_ (D_2L_R), generated by alternative splicing [17,18]. The difference is a 29-amino acid fragment insertion in the third intracellular loop (ICL3) of the D_2L_R. Although a large number of dimer complexes of D_2_R with other GPCRs have been extensively documented ((A_2a_-D_2_R; [19,20])(β2-D_2_R; [21])(CB1-D_2_R; [22,23])), this receptor can form homodimer complexes as well. It was first reported in 1996 by Ng et al. [24] that D_2L_R forms homodimers, as observed by co-immunoprecipitation (co-IP). Further evidence for homodimerization of both isoforms has been provided by studies using a wide variety of biochemical techniques such as co-IP, ligand binding [25], fluorescence resonance energy transfer (FRET) [26], bioluminescence resonance energy transfer (BRET) [27], single-molecule tracking [28], and protein–protein docking [29,30]. Furthermore, it has been suggested that the extent of dimerization is subtype-selective (D_2L_R > D_2S_R), suggesting a possible role for the 29-amino-acid fragment in ICL3 [31]. 

In order to better understand the crosstalk between dopamine receptors, the interface(s) should be considered from a molecular point of view. Different transmembrane (TM) regions of the D_2_R have been reported to be involved in the D_2_R homodimer interface. Incubation of D_2_R homodimers with peptides derived from the putative TM6 regions of the D_2_R resulted in dissociation of the dimer to the monomers [24,32]. On the other hand, successive deletion of TM domains of the D_2_R and cysteine cross-linking studies revealed that the most critical areas involved in the intermolecular hydrophobic interactions for dimerization resided in TM4 [33,34]. In addition, the TM4–TM5–TM4–TM5 and TM5–TM6–TM5–TM6 interfaces have been widely described to be involved in D_2_R hetero-oligomerization with other class A GPCRs [35,36,37,38,39,40]. In 2014, Guitart et al. [41] reported that dopamine D_1_ receptor (D_1_R) TM5- or TM6-derived single peptides were able to reduce D_1_R homodimerization. Likewise, a potential TM5–TM6–TM5–TM6 interface could be envisaged in the D_2_R homodimer. Collectively, these reported features support the hypothesis of multiple oligomerization interfaces [42], wherein GPCRs undergo multiple cycles of monomer and dimer formation with different interfaces. These interfaces can differ between homo- and heterodimerization processes of GPCRs. This concept of oligomerization of the D_2_R has also been confirmed by combined FRET and BRET assays, wherein at least four dopamine D_2_R monomers are closely located at the plasma membrane, suggesting higher-order oligomer formation [43,44].

Although it was first postulated that D_2_Rs form constitutive dimers or higher-order oligomers [34], increasing evidence supports the dynamic interconversion between monomers and dimers, suggesting transient dimer formation [28,42]. Recently, a lifetime of 0.5 s was determined for SNAP-tagged D_2L_R dimers using single-molecule sensitive total internal reflection fluorescence (TIRF) microscopy [31]. Whereas Tabor et al. detected transient D_2_R homodimer formation at 24 °C, Kasai et al. (2017) [28] performed single-molecule imaging at the physiological temperature of 37 °C, resulting in transient D_2_R dimer formation with a lifetime of 68 ms. Similar findings for temperature-dependent lifetimes of homodimer formation were also observed for other class A GPCRs [45,46,47]. 

The emerging evidence on transient dynamics of class A GPCR dimers, characterized by fast association and dissociation events, adds to the understanding of the complexity of receptor dimerization. Considering the dynamics and transient nature of D_2_R dimers, one might anticipate a functional relevance for alterations in the level of D_2_R dimerization. Indeed, an increase in D_2_R homodimer formation has been correlated with the pathophysiology of schizophrenia [48]. Therefore, targeting these D_2_R dimers might offer new information about the pathophysiology of diseases related to this GPCR dimer, potentially opening new therapeutic avenues. 

Within the concept of altering the level of dimerization or even oligomerization provoked by ligands, different screening methods have been implemented. For example, FRET has been used to monitor dose-dependent increases in the level of D_2S_R oligomerization by the agonist (−)-norpropylapomorphine [26]. Tabor et al. (2016) [31] used TIRF microscopy to investigate the effect of D_2_R agonists dopamine and quinpirole on the spatial and temporal organization of D_2_R dimer formation. These authors found that agonist stimulation at high concentrations (15 µM) seemed to prolong the lifetime of the D_2_R homodimer by a factor of ~1.5, whereas the neutral antagonist UH-232 (0.1 µM) did not alter the lifetime of the dimer.

To our knowledge, research on monovalent antagonist-mediated modulation of D_2_R dimerization is rather limited. The neutral UH-232 and 1,4-DAP have been tested, but no effect was observed [31]. In the present study, the modulating capacity of several clinically used D_2_R antagonists/inverse agonists on the level of D_2_R homodimerization or higher-order oligomerization was evaluated using complementation-based NanoLuciferase^®^ Binary Technology (NanoBiT^®^). In addition, an atomistic computational study of D_2_R conformational changes induced by specific D_2_R antagonists/inverse agonists and its relevance on D_2_R homodimerization has been performed using microsecond-length unbiased molecular dynamics (MDs) simulations.

## 2. Results

### 2.1. Pharmacological Properties of the D_2L_R Fusion Proteins

For the development of a complementation-based GPCR dimer targeting strategy, the D_2L_R was C-terminally fused to the small 1 kDa subunit (Small BiT, SmBiT) and to the large 18 kDa subunit (Large BiT, LgBiT) of NanoLuciferase. Upon interaction with D_2L_R monomers, the NanoLuciferase subunits were brought into close proximity and re-assembled spontaneously into a functional protein. To ensure that these modified D_2L_R fusion constructs retained functionality, we performed a G-protein coupling assay. We therefore cloned the mini-Gαi protein, corresponding to the engineered GTPase domain of the Gαi subunit fusion proteins, into the NanoBiT vectors with either LgBiT or SmBiT at their N-terminus. These mini-Gαi fusion proteins were transiently co-expressed with the corresponding (complementary) D_2L_R fusion constructs in HEK293T cells that were stimulated with the dopamine D_2_R agonist quinpirole (0.01 nM–10 µM). N-terminally tagged mini-Gαi proteins showed a concentration-dependent recruitment to the D_2L_R–SmBiT and D_2L_R–LgBiT fusion constructs (Figure 1). This demonstrated (i) that both receptor fusion constructs were expressed at the cell surface, (ii) that both receptor fusion constructs were responsive to ligand-induced activation, and (iii) that both receptor fusion constructs could still undergo a conformational change upon receptor modulation. Interestingly, the different construct combinations resulted in a dissimilar output in terms of sensitivity and signal-to-noise ratio, as published previously for the G-protein coupling assay with D_2_R [49]. Accordingly, pEC_50_ values for D_2L_R–LgBiT and SmBiT–mini-Gαi in comparison with D_2L_R–SmBiT and LgBiT–mini-Gαi deviated substantially (pEC_50_: 6.62 ± 0.02 and 7.65 ± 0.05, respectively). Although both D_2L_R fusion proteins can recruit mini-Gαi in a concentration-dependent manner and, thus, are functional, these observations further underscored the importance of testing several construct combinations when implementing systems like this for deducing EC_50_ values.

### 2.2. Targeting the Dopamine D_2L_R Homodimer using the NanoBiT Assay

To target D_2L_R homodimers in their native cell environment, the D_2L_R–LgBiT and D_2L_R–SmBiT fusion constructs were transiently transfected in HEK293T cells. This cell line was selected because of its high transient transfection efficiency as well as its rapid growth characteristics. More importantly, within a comparative study of four different cell lines frequently used for GPCR research, the HEK293 cell line showed the lowest expression (both amount and type) of GPCRs and could thereby serve as an appropriate cell model into which gene constructs of interest can be introduced [50]. Within this experimental setup, a clear luminescent signal was obtained when the D_2L_R–LgBiT and D_2L_R–SmBiT were co-expressed, indicating interaction of both receptors (Figure 2A). As negative controls, expression of the D_2L_R–LgBiT or D_2L_R–SmBiT separately only generated a signal that could be considered as background (i.e., seven- to ten-fold lower compared to the signal observed for the D_2L_R homodimer), as expected. As an additional negative control, we co-expressed the HaloTag–SmBiT construct, a fusion protein that is diffusively expressed throughout the cell. Again, a response not significantly (*p* > 0.05) different from background was detected (i.e., a five-fold lower signal was observed as compared to the signal provoked by the D_2L_R homodimer). Furthermore, from a screening of multiple GPCRs, the cannabinoid receptor 2 (CB2) was selected as a non-interacting partner for D_2L_R since no significant (*p* > 0.05) increase in luminescent signal was observed for the CB2–D_2L_R combination in direct comparison to the negative control D_2L_R–LgBiT with HaloTag–SmBiT. To our knowledge, no dimer formation of CB2 with D_2_R has been reported, in contrast to the CB1 for which dimerization with the D_2_R has been described [51]. Functionality of the CB2 constructs was demonstrated elsewhere [52]. In addition, the signal obtained for CB2–D_2L_R was significantly (four-fold) lower compared to that obtained for the D_2L_R–D_2L_R combination. The aforementioned results supported the utility of a NanoLuciferase complementation assay to differentiate between interacting (D_2L_R–D_2L_R) and non-interacting GPCRs (CB2–D_2L_R), when compared to background.

### 2.3. Antagonist-Dependent Modulation of the Level of D_2L_R Homodimer Formation

#### 2.3.1. Short-Term Effects

The short-term effect of the D_2_R antagonists haloperidol, spiperone, and clozapine on the level of dimerization was first evaluated on adherent HEK293T cells transiently transfected with D_2L_R–LgBiT and D_2L_R–SmBiT. Observed luminescent signals were corrected for solvent control, and the normalized relative luminescence units (RLU) were plotted against time (Figure 2B). A steeper drop in luminescent signal was observed when incubated for 1 h with spiperone (10 µM) compared to haloperidol or clozapine (Figure 2B). Although one should recognize the possible decay of the NanoGlo substrate, which was considered similar in all conditions, nevertheless, a clear difference in decrease in luminescent signal was observed when incubated with different antagonists (spiperone > haloperidol > clozapine).

#### 2.3.2. Long-Term Effects

For longer incubation time points, the capability of modulating the level of dimerization of the D_2_R antagonists haloperidol, spiperone, and clozapine was validated on cells in suspension. To circumvent fluctuations in the observed effect due to transfection variability, the obtained luminescent signal was normalized to the fluorescent signal obtained from the same amount of co-transfected Venus protein in all conditions. The normalized luminescent signal was measured after 10 min (Appendix A), 30 min (Appendix A), 4 h and 16 h of incubation with the D_2_R antagonists (Figure 2C,D). The effect of spiperone on the D_2L_R homodimer could be observed after 30 min and was sustained for up to 16 h of incubation. Spiperone reduced the level of D_2L_R dimerization by 40%–60%, depending on the time interval of incubation. This decrease in D_2L_R dimerization levels was only provoked upon incubation with a spiperone concentration ≥10 µM.

#### 2.3.3. Screening of a Broader Panel of D_2_R ligands

To investigate a possible class-dependent effect of D_2_R antagonists on the D_2L_R dimer, a broader panel of D_2_R ligands, including droperidol, spiperone, clozapine, olanzapine, risperidone, quinpirole, and haloperidol, was screened for their capacity to modulate the level of D_2L_R homodimer formation following long-term incubation (16 h). Of these, droperidol, clozapine, risperidone, and the D_2_R agonist quinpirole did not significantly (*p* > 0.05) modify the luminescent signal provoked by the dimer (Figure 3A). Haloperidol only slightly decreased the level of dimer formation (±30%). On the other hand, the D_2_R antagonist olanzapine clearly enhanced the luminescent signal by 45%. Finally again, the most significant effect was seen upon incubation with spiperone, with a clear reduction of 40–60%.

### 2.4. Validation of the Spiperone-Modulating Capacity on the D_2L_R Homodimer

Several experimental setups were implemented to validate the modulating capacity of spiperone on the D_2L_R dimer by investigating: (i) possible artifacts, (ii) expression levels of the D_2L_R, and (iii) the specificity of the effect of spiperone on the D_2L_R dimer. 

Firstly, to exclude that the observed effect was a result of possible artifacts such as toxicity, the possible influence of spiperone on the activity of native NanoLuciferase, transiently expressed in HEK293T cells, was investigated. Cells expressing the native luminescent enzyme were incubated for different time points with 10 µM of the antagonists. No impact on luciferase activity was observed (Appendix A). 

Secondly, to rule out a possible role for spiperone on the expression level of the D_2L_R, the receptor was fused to yellow fluorescent protein (YFP) and HEK293T cells transiently transfected with the fusion construct, and they were incubated with 10 µM of the D_2_R antagonists haloperidol, clozapine, and spiperone (Figure 3B). Incubation with these D_2_R antagonists did not cause any significant alteration in the level of fluorescent signal after 16 h of incubation. Similarly, a western blot experiment under reducing conditions was conducted to analyze the expression of the fusion proteins D_2L_R–SmBiT and D_2L_R–LgBiT in both cells that had been and had not been incubated with the antagonists (Figure 3C). The aim of this experiment was merely to evaluate whether there was an impact on D_2L_R expression. After normalization to tubulin as a housekeeping protein, a 14% decrease of D_2L_R fusion protein expression was observed in cells treated with spiperone, compared to the solvent-treated control. Under these (reducing) conditions, no clear bands of D_2L_R dimers or higher oligomers could be observed, which might be explained by the fact that lower densities of receptors in the plasma membrane could conceivably reduce the proportion of receptors forming dimers, as reported before [53].

Finally, the specificity of the effect of spiperone on the D_2L_R homodimer was evaluated by examining its effect on another well-studied GPCR dimer, namely the adenosine A_2a_ receptor–D_2_R dimer [19,20,54]. We therefore co-expressed A_2a_–LgBiT and D_2L_R–SmBiT in HEK293T cells that were treated with 10 µM spiperone for 16 h (Figure 3D). No significant effect (*p* > 0.05) was observed on the level of A_2a_–D_2L_R dimer formation, lending further support to the specificity of the effect of spiperone on the D_2L_R homodimer.

### 2.5. Spiperone and Clozapine Achieve Stable Binding Poses in D_2_R during Molecular Dynamics Simulations

In order to comprehend how spiperone might reduce D_2_R homodimerization relative to clozapine at the molecular level, we first docked each ligand into the crystal structure of D_2_R (PDB id: 6CM4) [55] and then performed unbiased molecular dynamics (MDs) simulations to allow for ligand-induced conformational changes to occur in the monomeric receptor. During respective time periods of 3 µs, both spiperone and clozapine achieved stable binding poses (Figure 4A) despite some initial conformational changes in both ligands, as might be expected (Appendix A). Specifically, clozapine and spiperone achieved stable bound conformations from 0.4 and 1.8 μs onwards, respectively, where root mean square deviation (RMSD) from their final conformations remained <3.0 Å (average of 1.5 Å ± 0.5 S.D. for clozapine and 1.9 Å ± 1.0 S.D. for spiperone). The relative higher conformational fluctuation observed with spiperone can be attributed to its greater flexibility, mainly due to its central alkyl chain. Despite clozapine and spiperone reaching stable binding poses at different times, in both cases the D_2_R monomer presented little conformational change of its backbone, with final values of 2.5 Å and 2.2 Å, respectively (Appendix A). In the original crystal structure, residues in close contact with co-crystallized risperidone [54] (<3.5 Å) were located on extracellular loop 1 (ECL1), TM3, TM5, and TM6 (Table 1 and Appendix A). In terms of the protein–ligand interactions in common between risperidone, spiperone, and clozapine, the most prominent was an electrostatic interaction between the protonated ligand amine group and Asp114^3.32^ (superscript numbers refer to the Ballesteros and Weinstein generic numbering scheme [56], which includes relative TM helix location) on TM3, which was maintained over respective MD simulations. Other common interactions, which occurred once each ligand found its stable binding pose, included contacts with residues on TM3, TM5, and extracellular loop 2 (ECL2): Val115^3.33^, Ile184^ECL2^, and Ser193^5.42^ (Figure 4A and Table 1). However, clozapine established several distinct contacts with residues on TM5 and TM6. On the other hand, spiperone was frequently in contact with residues on TM2 and TM3 (Table 1). These different residues in contact with clozapine and spiperone demonstrated that their binding poses were quite different (Figure 4A and Table 1).

### 2.6. Spiperone and Clozapine Select for Different Sidechain Conformations in D_2_R TM5 and TM6

To ascertain the most important conformational changes selected by the stable binding poses of clozapine and spiperone in D_2_R, we carried out residue-level analyses of the monomeric MD simulations. No significant conformational differences between systems were observed in any residues located on TM helices, except for TM5 and TM6. Specifically, neighboring residues Tyr199^5.48^ and Phe390^6.52^ showed different χ1 dihedral angle conformations with different bound antagonists. In general, for these two aromatic residues, two different pseudo-stable conformations can be observed in our MD simulations, a cis and a trans χ1 dihedral angle of 300° and 180°, respectively (Figure 4B). The D_2_R crystal structure presented cis conformations for both Tyr199^5.48^ and Phe390^6.52^, which underwent conformational changes to trans more frequently when clozapine was bound than with spiperone (Appendix A). Specifically, considering only time periods where clozapine and spiperone presented stable binding poses (from 0.4 and 1.8 μs onwards, respectively) clozapine preferentially selected for Tyr199^5.48^ and Phe390^6.52^ χ1 trans conformations 99% of the time. This “double” χ1 trans conformation led to an outward orientation (towards the membrane) for Tyr199^5.48^ and Phe390^6.52^, which potentially may encourage protein–protein interactions through the formation of aromatic contacts, which are known to be important (Figure 4B) [57]. Conversely, spiperone induced rapid fluctuations between χ1 cis and trans conformations of Tyr199^5.48^, with the cis selected 25% of the time, whereas the cis conformation of Phe390^6.52^ was exclusively maintained. The χ1 cis conformation of Tyr199^5.48^ and Phe390^6.52^ oriented them in a more inward position, away from the membrane, which may conceivably discourage protein–protein interactions (Figure 4B).

### 2.7. Aromatic Interactions Stabilize D_2_R Homodimer Model Interface during MD Simulation

In order to probe how a D_2_R homodimer might be affected by sidechain conformational changes in TM5 and TM6, a D_2_R homodimer without bound antagonist was modeled from the original crystal structure with a TM5–TM6–TM5–TM6 interface (in line with experimental evidence by Pulido, 2018 [32]) by protein–protein docking. This resulted in a D_2_R homodimer with a highly favorable interface docking score of −9.7 (on a scale of 0 to −10, where lower than −5.0 was considered satisfactory ([58]); see Methods). This model was subjected to an MD simulation of 3 µs to investigate homodimer physical stability and receptor conformational changes in individual protomers (Figure 4C). During this MD simulation, the TM5–TM6–TM5–TM6 interface remained intact, according to a consistently close interaction distance (Appendix A) between TM5/TM6 helices of each protomer. In the process, participating helices experienced a moderate backbone conformational change of 3.2 Å in order to enhance mutual binding, obtaining an average interaction energy of −11.7 kcal/mol (±3.2 S.D.) between protomers (Appendix A). Furthermore, from an analysis of individual protomers in the homodimer, it can be observed that one protomer underwent slightly more backbone conformational changes than the others during the second half of MD (average RMSDs of 2.9 Å and 2.3 Å, respectively (Appendix A)). To ascertain the relevance of interactions involved in the TM5–TM6–TM5–TM6 homodimer interface, we performed a conformational and energetic analysis of specific residues on TM5 and TM6. Interestingly, the D_2_R protomer whose backbone remained relatively unchanged rapidly selected for the cis to trans conformational change of Tyr199^5.48^ and Phe390^6.52^ χ1 dihedral angles (Figure 4C), which occurred at 94% and 90% of the total MD simulation time, respectively (Appendix A). In addition, a rapid cis to trans conformational change of Phe390^6.52^ χ1 dihedral angle was observed in the other protomer (occurring at 93% of total time). However, in this second protomer, Tyr199^5.48^ presented no significant conformational change and remained in the cis conformation (Appendix A). As shown in Figure 4C, the outward conformations achieved by Tyr199^5.48^ and Phe390^6.52^ in the homodimer enabled an aromatic interaction network to form, as well as transient H-bond formation between Tyr199^5.48^ sidechains of both protomers (H-bond occupancy of 4%). As a result, the average minimum distance between Tyr199^5.48^/Phe390^6.52^ residues of each protomer was 5.5 Å (±1.5 S.D.) (Appendix A). From an energetic point of view, alanine scanning of Tyr199^5.48^ and Phe390^6.52^ confirmed the relevance of these residues in the D_2_R homodimer interface. Removal of these aromatic interactions (by alanine mutation) resulted in a less favorable average interface energy of −8.6 kcal/mol (±2.8 S.D.), which suggested this aromatic interaction network contributed an average of −3.1 kcal/mol to the homodimer interface.

### 2.8. D_2L_R Oligomerization

HEK293T cells were co-transfected with 400 ng D_2L_R–SmBiT, D_2L_R–LgBiT, and increasing DNA concentrations of native D_2L_R (0–600 ng) (Figure 5). Co-expression of native D_2L_R did not circumvent or attenuate the complemented luminescent signal, but in fact it stimulated the D_2L_R oligomerization (Figure 5A) in an expression-dependent manner. To rule out that crowding of GPCRs on the membrane or nonspecific aggregation would result in trivial complementation of the NanoBiT proteins, the muscarinic M1 receptor was co-transfected instead of the native D_2L_R (Figure 5B). Co-expression of the M1 receptor did not modify the luminescent signal in a significant manner. Furthermore, also in the presence of more D_2L_Rs, spiperone still had an impact: upon treatment, the increase in oligomerization by increasing amounts of native D_2L_R was less pronounced (Figure 5C). Specifically, when comparing the experimental setup of HEK293T cells transiently expressing D_2L_R–LgBiT and D_2L_R–SmBiT with the same setup but with high levels of co-transfected native D_2L_R (4:4:6) (Figure 5A), a significant twenty-fold increase in luminescent signal was observed for the latter. On the other hand, when treated with spiperone, only a five-fold difference between the same two experimental setups could be observed (Figure 5C). 

## 3. Discussion

Over the past two decades, a growing body of evidence suggests that GPCRs are able to form dimers and/or even higher-order oligomers [59]. Because these GPCRs are involved in many physiological processes, these dimeric or oligomeric GPCR complexes are not only of paramount importance for possible alterations in signaling cascades, compared to their monomers, but also for their association with debilitating diseases.

Interestingly, a significant increase of D_2_R dimerization has already been observed in post-mortem striatal tissue of schizophrenia patients [48]. Concerning the dimer formation of D_2_Rs, as well as other class A subfamily GPCRs [45,46], clear evidence for the transient characteristics of the dimer formation has been provided by single-molecule tracking studies [31], with a lifetime of 68 ms being assigned to the D_2_R dimer [28,42]. Although a lot of knowledge has been gathered concerning dimer formation of the D_2_R, key questions still remain unanswered. For example, different D_2_R dimeric interfaces have been proposed [29,33], as well as a hypothesis of multiple oligomerization interfaces [42]. Nevertheless, a recent interest has arisen in the establishment of a D_2_R TM5–TM6–TM5–TM6 dimeric interface [32,40,41]. In our present study we have addressed the computational reliability of this dimeric interface and its implication in the D_2_R homodimer by means of computational techniques including microsecond-length MD simulations. In addition, targeting GPCR dimers with ligands or selective chemical tools may elaborate the signaling behavior of dimers as well as their tendency or preference towards GPCR–GPCR interactions. Nonetheless, this topic of ligand-induced modulation of GPCR dimers has been much debated [60,61,62], with both arguing for and against. In the current study, we further elaborate on this topic and demonstrate the ability of spiperone to alter the dynamic equilibrium between D_2L_R monomers and dimers, with a clear preference towards monomers.

The Nanoluc^®^ Binary Technology (NanoBiT^®^), developed by Promega in 2016 [63], proved to be an interesting tool to study D_2_R dimers. In contrast to other complementation-based assays, NanoBiT^®^ offers the great advantage of being reversible, which gives opportunity to look into detail on the kinetics of GPCR interactions. Importantly, since this system requires the fusion of LgBiT or SmBiT to the GPCR of interest, the functionality of D_2L_R fusion proteins was demonstrated by mini-Gαi protein recruitment to both receptors upon stimulation with the D_2_R agonist quinpirole. 

Using this experimental HEK293T-based design, we screened six different D_2_R antagonists and one agonist for their ability to modulate the level of D_2L_R dimer formation. This panel of ligands compromises droperidol, spiperone, clozapine, olanzapine, risperidone, quinpirole, and haloperidol. Although several of the aforementioned ligands have previously been classified as antagonists, one should keep in mind that their inverse agonist capacity has now been recognized [64,65,66,67]. Of those, the D_2_R antagonist/inverse agonist spiperone could significantly decrease the level of D_2L_R dimers by 40%–60% in real-time and after long-term (up to 16 h) incubation. Another D_2_R antagonist, haloperidol, also modulates the level of D_2L_R dimerization, but in a less significant manner (±30%). In contrast, the D_2_R antagonist olanzapine significantly increases the level of D_2L_R dimer formation by ±45%. Furthermore, a class-dependent effect between the butyrophenones (haloperidol, spiperone, and droperidol) and atypical antipsychotics (clozapine, risperidone, and olanzapine) could not be distinguished. For the D_2_R agonist quinpirole, only a minor increase in luminescent signal provoked by D_2L_R dimer formation could be observed. Although it was demonstrated that agonist addition (i.e., dopamine and quinpirole) stabilized the formation of D_2_R dimers by a factor of 1.5 in a total measure time of 400 ms by single-molecule tracking [28], we might conclude from this study that this modulating effect does not significantly hold true for long-term effects. Nevertheless, one should keep in mind that findings might differ due to diverse experimental assay setups as well.

In order to further examine the modulating capacity of spiperone on the D_2L_R dimer, we performed screenings towards incubation time (real-time vs long-term effects), expression levels of D_2L_R, and the specificity of the effect on the D_2L_R dimer. From this, we can conclude that a decrease in the level of D_2L_R dimerization could readily be observed after approximately 30 min and was still detected after long-term incubation up to 16 h. These data are in agreement with findings for the dopamine D_3_R homodimer, for which similar effects were observed after 16 h treatment with spiperone [68]. As a control, we examined whether spiperone altered expression levels of the D_2L_R, which could cause a decrease in luminescent signal. This possibility was ruled out by both western blot analysis and the fluorescence analysis of a D_2_R–YFP fusion protein, expressed in cells that were or were not treated with spiperone for 16 h.

Additionally, to ensure the specificity of the effect of spiperone on the D_2_R homodimer, the same experimental set-up with another GPCR dimer was investigated. For this, we selected the well-studied adenosine A_2a_ receptor (A_2a_) and D_2_R dimer since many research groups have reported on: (i) the formation of the dimer by several techniques such as BRET and FRET [20,69] and protein complementation assays [70], (ii) specific dimer characteristics regarding signaling pathways of the A_2a_–D_2_R dimer [71,72], (iii) the dimer interface [73], and (iv) allosteric mechanisms [54], among others. Importantly, the fact that several studies have linked this dimer to Parkinson’s disease [74,75,76,77] lends support to the relevance of research within this field. Nevertheless, to the best of our knowledge, the modulating capacity of the D_2_R antagonist spiperone on the level of A_2a_–D_2_R dimer formation has not been investigated yet. Overall, the effect of spiperone on the A_2a_–D_2L_R dimer was evaluated by treatment with 10 µM spiperone for 16 h. However, this did not have a significant effect on the level of A_2a_–D_2L_R dimer formation. Thus, since the spiperone-modulating capacity does not hold true for all D_2_R dimer complexes, this effect might be specific for the D_2L_R homodimer or oligomer.

Computational techniques such as MD simulations have shown promise for studying GPCRs, such as D_2_R, and their mechanisms of signaling transmission at the atomic level [30]. From a computational point of view, in our study we observed noticeable differences between the orthosteric binding poses of spiperone and clozapine in a D_2_R monomer, which select for different sidechain conformations of Tyr199^5.48^ and Phe390^6.52^ on TM5 and TM6, respectively. Interestingly, the inward conformations adopted by Tyr199^5.48^ and Phe390^6.52^ when spiperone is bound differ from the outward conformations induced by clozapine, which are also favored in the modeled D_2_R homodimer. In this study we have observed aromatic interactions between Tyr199^5.48^ and Phe390^6.52^, as well as occasional H-bonding between Tyr199^5.48^, of both protomers in a model D_2_R homodimer, which could be indicative of the relevance of these two residues in the establishment of a TM5–TM6–TM5–TM6 interface and their role in the homodimerization process. In addition, our D_2_R homodimer model with a TM5–TM6–TM5–TM6 interface, in accordance with a previously published D_2_R–mGlu5 heterodimer model presented by Qian et al. (2018) [40], is physically stable over microsecond-length MD simulations. In addition to this homodimeric interface, it has been widely described that D_2_R heteromerizes through a TM4–TM5–TM4–TM5 interface with other class A GPCRs, such as A_2a_ and AT1 receptors [35,36,37,38,39]. Therefore, our results raise questions about the oligomerization interfaces D_2_R may form. In our present study we observe that conformational changes specifically occurring in TM5 and TM6, resulting from bound spiperone and involving inward Tyr199^5.48^ and Phe390^6.52^ sidechain conformations, may alter the TM5–TM6–TM5–TM6 D_2_R homodimer interface. This fact may explain the results observed in our experimental approach where spiperone specifically reduces levels of the D_2_R homodimer, while having no significant effect on A_2a_–D_2_R heterodimer formation. Altogether, these results indicate that the interfaces involved in homodimerization of D_2_R may differ from the interfaces involved in heterodimerization processes with class A GPCRs, which could also differ between different GPCR classes, in agreement with the hypothesis of multiple oligomerization interfaces presented by Kaisai et al. (2014) [42].

Finally, D_2L_R oligomerization was investigated in a similar experimental design, using the NanoBiT^®^ assay. Although it was first postulated that co-transfection of the native D_2L_R would attenuate the luminescent signal provoked by D_2L_R dimer formation by competing for interaction with the D_2L_R–SmBiT and D_2L_R –LgBiT fusion proteins, the opposite was observed. Oligomerization of the D_2L_R appeared to be concentration-dependent, with higher expression levels of native D_2_R provoking complementation of the fusion proteins because of the close proximity to their corresponding receptors, suggesting stimulation of the organization as higher-order oligomers. The fact that the same outcome was not observed when co-expressing increasing amounts of the muscarinic M1 receptor confirms that this effect was not due to nonspecific aggregation or crowding of GPCRs. The finding of an increased D_2_R homo-oligomerization with higher levels of expression is in agreement with literature [26,31,43,44,78] and has been reported for other dopamine receptors as well [68,79]. In addition, the effect of spiperone was evaluated on higher expression levels of D_2L_R as well. Also here we demonstrated that spiperone reduces the level of D_2L_R–D_2L_R interactions. Rather than a twenty-fold increase of luminescent signal resulting from higher D_2L_R expression levels, pre-treatment with spiperone only resulted in a five-fold increase. To conclude, higher expression levels stimulate D_2L_R–D_2L_R interaction, suggesting oligomerization. Also at these higher expression levels, spiperone still exerts a negative impact on D_2L_R–D_2L_R interactions. Consistent with this concept, one might speculate that spiperone could exert different pharmacological properties in different areas of the brain, in co-relation with the expression level of D_2L_R.

Interestingly, Ng et al. (1996) [24] postulated that spiperone favors binding to the monomer over the dimer, whereas risperidone binds to monomers as well as dimers. In light of our findings, one might hypothesize that spiperone does not necessarily favor binding to the monomers, but simply reduces the number of dimers, as observed in this study.

On the contrary, Armstrong et al. (2001) [25] reported quite opposite data obtained from ligand binding experiments. These authors proposed a model wherein D_2_Rs can form dimeric units with two orthosteric binding sites for two equivalents, which allows allosteric cooperativity. From experimental data, it was suggested that the first and second equivalent of [^3^H]spiperone only exerted limited cooperativity between the dimer units, in the absence or presence of sodium ions. On the other hand, [^3^H]raclopride seems to prefer binding to monomeric units because of an observed negative cooperative effect on the binding of the second equivalent upon binding of the first equivalent, which results in a reduced affinity of the second site of the dimer for [^3^H]raclopride. Within the mindset of this proposed model by Armstrong et al. [25], [^3^H]spiperone binds to the D_2_R dimer, and although no negative effect on affinity of both binding sites due to cooperativity was observed by the authors, from our data we can suggest that conformational changes within the dimer upon spiperone binding might lead to dissociation of the dimer to its monomers. 

Interestingly, a similar destabilizing effect of spiperone on D_3_R oligomeric complexes was reported by Marsango et al. (2017) [68]. Using a spatial intensity distribution analysis (SpIDA) method, the antipsychotics spiperone and haloperidol reduced the level of D_3_R dimerization in a ligand-dependent manner. Moreover, this effect could be reversed upon ligand washout. Since the D_3_ and D_2_ receptors are highly homologous and show a sequence identity of 78% [80], it might not be surprising that certain ligands modulate these receptors in a similar way.

Although the development of the reversible complementation-based NanoBiT assay allows the screening and discovery of ligands that could modulate the level of dimerization, this technique does not provide information about the dynamics of the D_2_R dimers or oligomers at the single molecule level. To allow visualization and tracking in real-time of the influence of spiperone on a D_2_R dimer in the membrane of living cells, techniques such as single-molecule sensitive total internal reflection fluorescence microscopy (TIRF-M) are recommended. Thus, based on the present understanding, further research to study the effect of the D_2_R antagonist spiperone on the D_2_R homodimer in detail is required.

## 4. Materials and Methods

### 4.1. Chemicals and Reagents

Dulbecco’s modified Eagle’s medium (DMEM) supplemented with GlutaMAX, Opti-MEM^®^ I reduced serum medium and Gibco™ Penicillin-Streptomycin (10,000 U/mL), Hank’s balanced salt solution (HBSS), Phusion high-fidelity (HF) PCR master mix with HF buffer, and T4 DNA ligase were purchased from Thermo Fisher Scientific (Pittsburg, PA, USA). Fetal bovine serum (FBS) was purchased from Biochrom, now part of Merck (Merck KGaA, Darmstadt Germany). Phosphate buffered saline (PBS) was procured from Lonza (Lonza, Walkersville, MD, USA). Transient mammalian cell transfection reagent polyethylenimine (PEI), poly-D-lysine, carbenicillin, Tween 20, and DMSO suitable for cell culture were purchased from Sigma-Aldrich (Steinheim, Germany). D_2_R antagonists spiperone, clozapine, and haloperidol were purchased from Tocris Bioscience (Bio-techne, Abingdon, UK). The Nano-Glo^®^ Liv e Cell reagent and the GoTaq^®^ DNA polymerase were from Promega (Madison, WI, USA). Primers were synthesized by Eurofins Genomics (Ebersberg, Germany). Restriction enzymes HindIII and EcoRI were from New England Biolabs (NEB, Massachusetts, US). E.Z.N.A.^®^ MicroElute Gel extraction kit, E.Z.N.A.^®^ MicroElute Cycle-Pure kit and E.Z.N.A. plasmid DNA Mini/Midi kit were from VWR International (Radnor, PA, USA). GelRed was purchased from Biotium (Fremont, CA, USA). Luria Bertani broth and agar were procured from Lab M (Heywood, Bury, UK).

### 4.2. Cloning of the Dopamine D_2_R into the NanoBiT^®^ plasmids

The human D_2L_R (NM_000795.3) was cloned into the NanoBiT^®^ vectors (NB MCS1 and NB MCS2), which were kindly provided by Promega (Madison, WI, USA). The NanoBiT^®^ constructs express a small subunit of the NanoLuciferase of 1 kDa (Small BiT, SmBiT) and a large subunit of 18 kDa (Large BiT, LgBiT). The D_2L_R was cloned into the NanoBiT^®^ vectors prior to a 15 amino acid encoding sequence, linking it to the SmBiT or LgBiT fragment, by performing a PCR reaction with primers containing the specific restriction enzyme sites (Table 1). The PCR reaction was performed with an MJ Research PTC-200 Thermal Cycler (GMI, Minnesota, USA), in a three-step manner: initial denaturation (98 °C, 30 s), denaturation (98 °C, 10 s), annealing (Tm, 35 s), extension (72 °C, 42 s), and final extension (72 °C, 5 min), for 30 cycles. PCR products were run on a 0.1% agarose gel and purified with a MicroElute Gel extraction kit to remove parental DNA. After digestion with the specific restriction enzymes for 3 h at 37 °C, the PCR product and the NanoBiT^®^ vectors were purified with a MicroElute Cycle-Pure kit and a MicroElute Gel extraction kit, respectively. Following ligation using T4 DNA ligase for 1 h at room temperature, the ligated product was transformed into a competent MC1061 *Escherichia coli* strain. After plating on carbenicillin-containing agar, resistant colonies were screened for the presence of the insert by Colony PCR with *Taq* polymerase and subsequent restriction digest. Coding sequences were verified by Sanger sequencing (Eurofins Genomics, Ebersberg, Germany).

As a control, the cDNA coding the human A_2a_ receptor (A_2a_), a kind gift from F. Ciruela (Unitat de Farmacologia, Barcelona, Spain), was fused to SmBiT and LgBiT in a similar way as for the D_2L_R (Table 2). In addition, cannabinoid receptor 2 (CB2) fusion constructs, CB2–LgBiT and CB2–SmBiT, were developed by performing a PCR reaction on the human CB2 coding sequence (as described previously by our research group) [52].

### 4.3. Cell Culture

#### 4.3.1. Expression in HEK293T Cells

Human Embryonic Kidney 293T (HEK293T) (American Type Culture Collection (ATCC), Manassas, Virginia, USA) cells were maintained in DMEM supplemented with 10% fetal bovine serum (FBS), 100 µg/mL streptomycin, and 100 IU/L penicillin in a controlled environment (37 °C, 98% humidity, 5% CO_2_). Prior to transfection, cells were cultured in 6-well plates at a density of 3 × 10^5^ cells/well in 2 mL DMEM + 10% FBS. To ensure low expression levels of GPCRs, only 200 ng of each GPCR fused to a luminescent protein fragment was transiently transfected using PEI transfection reagent in DMEM supplemented with 2% FBS. After 5 h of incubation with the transfection mixture, the medium was refreshed with DMEM + 10% FBS.

#### 4.3.2. Cell Preparation for Dimerization Assay with HEK293T Cells in Suspension

Forty-eight hours after transfection, cells were washed twice with PBS, scraped, and centrifuged for 5 min at 1000× *g*. A bicinchoninic acid assay (BCA) was conducted on an aliquot of the transfected cells in HBSS buffer, and all protein concentrations were measured. The cell suspensions were diluted to bring them all to a density corresponding to a measured protein concentration of 600 ng/µL. For the dimerization assay, the Nano-Glo Live Cell reagent, a non-lytic detection reagent containing furimazine substrate, was 20× diluted using Nano-Glo Live Cell System (LCS) dilution buffer, and 25 µL was added to each 96-well containing 100 µL cell suspension. End-point fluorescence or luminescence was measured with the ClarioSTAR (BMG LABTECH) in a black and white 96-well plate, respectively.

#### 4.3.3. Cell Preparation for Dimerization Assay with Adherent HEK293T Cells

Twelve hours after transfection, cells were reseeded in poly-D-lysine-pretreated white 96-well plates at 0.5 × 10^5^ cells/well. The next day, cells were washed twice with Opti-MEM^®^ and 100 µL of the reduced serum medium was added to each well. First, 25 µL of the Nano-Glo Live Cell reagent was added, followed by an incubation of 15 min, monitored by the Tristar (as described previously [52]). Afterwards, 10 µL of solvent control (blank sample, DMSO ≤ 0.1%) or ligand was added to obtain a final concentration of 10 µM. The read-out was performed immediately upon treatment and monitored for 1 h at room temperature by the TriStar^2^ LB 942 multimode microplate reader controlled by ICE software (Berthold Technologies GmbH & Co., Bad Wildbad, Germany).

#### 4.3.4. Fluorescence Normalization and Signal-To-Noise Ratio 

To circumvent fluctuations in signal resulting from varying transfection efficiencies, a constant amount of a plasmid encoding the fluorescent protein Venus (10% of total DNA transfected) was co-transfected in all conditions. Luminescence data were normalized for the measured fluorescent signal.

As a negative control, the protein fragment SmBiT of the luminescence-based assays, not fused to a receptor but to the HaloTag, was implemented. The luminescent/fluorescent signal obtained for this condition (co-transfected with, e.g., D_2L_R–LgBiT) was considered as background and, consequently, a signal-to-noise ratio could be derived. 

### 4.4. NanoBiT^®^-Based Validation of the Functionality of D_2L_R Luminescent Fusion Proteins by mini-G_αi_ Protein-Mediated Signaling 

The plasmid encoding the mini-Gαi protein was kindly provided by the lab of Dr. A. Chevigné (LIH Luxembourg Institute of Health, Luxembourg). The construct was PCR-amplified using synthesized primers (Forward: 5′ ACTCAAGAATTCAATGATCGAGAAGCAGCTGCAG 3′ and Reverse: 5′ ACTCAAGAATTCTCAGAACAGGCCGCAGTCTCTC 3′) and subcloned into the NanoBiT^®^ constructs expressing LgBiT and SmBiT using *Eco*RI restriction sites flanked at both sites. Sequences were verified by Sanger sequencing.

HEK293T cells were seeded in 6-well plates at a cell density of 5 × 10^5^ cells/well. The next day, cells were transiently transfected with 1.5 µg of each construct (D_2L_R–LgBiT and SmBiT–mini-Gαi or D_2L_R–SmBiT and LgBiT–mini-Gαi) using FuGENE^®^ HD transfection reagent (Promega) according to the manufacturer’s instructions. For reseeding the cells in white 96-well plates, as well as monitoring of the luminescent signal, the same procedure was followed as described in ‘Cell preparation for dimerization assay with adherent HEK293T cells’. On the fourth day, cells were treated with quinpirole (0.01 nM–10 µM) to evoke mini-Gαi protein recruitment to the D_2L_R. 

### 4.5. Detection of the Expression Levels of D_2L_R Dimers by Western Blot

Western blot analysis was executed as previously described [81], with some minor adaptations. The day before transfection, HEK293T cells were seeded in 10-cm dishes at a density of 3 × 10^6^ cells/well. PEI-mediated transient transfection was performed with plasmids encoding D_2L_R–SmBiT and D_2L_R–LgBiT, each present at 2 µg per dish. The next day, cells were treated with 10 µM haloperidol, spiperone, clozapine, or solvent control for 16 h at 37 °C. On the fourth day, cells were washed two times with PBS, harvested, and lysed using Polytron homogenizer for two 10 s periods in ice-cold PBS buffer. Membrane pellets were obtained by centrifugation at maximum speed for 25 min at 4 °C and dissolving in RIPA buffer (150 mM NaCl; 50 mM Tris HCl, pH 7.5; 1% NP-40; 0.5% deoxycholic acid; supplemented with fresh protease inhibitors: 5 µg/mL aprotinin, 0.4 mg/mL pefabloc and 10 mM β-glycerol-phosphate disodium salt pentahydrate, and 10 μg/mL leupeptin). The membrane pellets were rotated for 1 h at 4 °C, followed by a centrifugation for 20 min at maximum speed. Next, the BCA method was performed on the supernatant to quantify the protein levels, with bovine serum albumin dilutions as the standard. Cell lysates (50 µg) were heated in Laemmli buffer supplemented with 10% β-mercaptoethanol and 5% bromophenol blue for 10 min at 37 °C. Proteins were separated via a 10% SDS-PAGE for 1 h at 100V and transferred to a polyvinylidene difluoride membrane. Membranes were blocked with blocking buffer (LI-COR Biosciences, Lincoln, NE, USA) for 1 h at RT and incubated with rabbit anti-D_2_R antibody (RRID: AB_2571596) (Frontier Institute, Hokkaido, Japan) overnight at 4 °C, followed by three washing steps with PBS + 0.05% Tween 20. Afterwards, blots were incubated for 1 h in the dark with goat anti-rabbit IRDye680 LT (1/10,000) (cat. no. 926–68021, LI-COR Biosciences, Lincoln, NE, USA) at RT. Equal loading of all conditions was assessed by normalization by the levels of the constitutively expressed neuronal marker tubulin with the monoclonal anti-α-tubulin antibody (cat. no. T5168, Sigma Aldrich, Steinheim, Germany). After incubation with the primary antibody for 1 h, followed by three washing steps with PBS + 0.05% Tween 20, blots were incubated for 1 h in the dark with the Alexa Fluor^®^ goat anti-mouse secondary antibody (cat. no. A-11001, Invitrogen, Carlsbad, CA, USA). Blots were visualized with the Odyssey^®^ Infrared Imaging system (IGDR, Rennes, France) and quantified by ImageJ software (NIH, Bethesda, MD, USA). 

### 4.6. Data Analysis

Concentration-response histograms were calculated after correction for the fluorescent signal measured in the same well to compensate for transfection variability. Statistics were performed using the non-parametric (Kruskal–Wallis) one-way ANOVA, followed by post hoc (Dunn’s multiple comparison test) analysis to detect statistical differences amongst groups (*p* < 0.05) by the GraphPad Prism software (San Diego, CA, USA).

Curve-fitting of concentration−response curves of the mini-Gαi coupling to the D_2L_R via a nonlinear regression model (variable slope, four parameters) was employed to determine pEC_50_ values (a measure of potency). The mean area under the curve (AUC) ± standard error of mean (SEM) was calculated, with a total of 12 replicates for each data point.

### 4.7. Computational Modeling

A previously published D_2_R model [40], based on human D_2_R crystal structure (PDB id: 6CM4) [55], was generated using CHIMERA v1.11.2 [82] software (San Francisco, CA, USA) by adding missing residues and converting the crystal mutated residues back to wild-type. In addition, co-crystallized risperidone and endolysin fusion protein were removed from the D_2_R structure. This D_2_R model was used as initial conformation for construction of three different molecular systems: (i) spiperone-bound D_2_R monomer, (ii) clozapine-bound D_2_R monomer, and (iii) D_2_R homodimer without bound antagonist. Coordinates for clozapine and spiperone were downloaded from PubChem [83]. AUTODOCK v4.2 [84] software (La Jolla, CA, USA) was used to dock clozapine and spiperone into the monomeric D_2_R model. The selected docked conformation of each ligand in the receptor represented the top hit identified by best predicted affinity in the largest docking cluster. For construction of the D_2_R homodimer model, where two protomers of D_2_R interacted via a symmetrical TM5–TM6–TM5–TM6 interface, two D_2_R monomers without bound antagonist were initially superimposed onto respective protomers of the μ-opioid receptor homodimer crystal structure (PDB id: 4DKL) [85]. The D_2_R homodimer model was then submitted to the ROSIE Web server [58] for protein–protein docking using default parameters. The best docked homodimer structure was identified by two factors: best interface score (“I_sc”) and best membrane-compatible orientation. The D_2_R monomer, with bound spiperone or clozapine, and D_2_R homodimer without bound antagonist complexes were energy minimized without restraints with CHIMERA [82] in the AMBER-14SB force-field [86] to optimize protein–ligand or protein–protein interactions, respectively.

### 4.8. Molecular Dynamic (MD) Simulations

D_2_R monomer, with bound spiperone or clozapine, and D_2_R homodimer without bound antagonist complexes were embedded separately into a 1-Palmitoyl-2-oleoylphosphatidylcholine (POPC) membrane and solvated with TIP3P water molecules using the CHARMM-GUI web-based interface [87]. Complexes were oriented in the membrane according to the OPM database [88] entry of D_2_R crystal structure (PDB id: 6CM4) [55] or μ-opioid receptor homodimer crystal structure (PDB id: 4DKL) [85] for monomer and homodimer models, respectively. Charge-neutralizing ions (0.15 M KCl) were introduced into each system. Parameters were automatically generated by CHARMM-GUI [87]. Membrane, water, and protein parameters were generated according to the CHARMM36 force-field [89], whereas spiperone and clozapine parameters were generated according to CGenFF v1.0.0 [90]. Molecular dynamics (MDs) simulations of D_2_R monomer, with bound spiperone or clozapine, and D_2_R homodimer were performed using the CHARMM36 force-field [89] with ACEMD [91] on specialized GPU-computer hardware (Stanmore, Middlesex, UK). Each system was equilibrated for 28 ns at 300 K and 1 atm, with positional harmonic restraints on protein heavy atoms progressively released over the first 8 ns of equilibration and then continued without constraints. After equilibration, monomer and homodimer models were subjected to unbiased continuous production runs under the same conditions for 3 µs.

### 4.9. MD Simulation Analysis

Analysis of MD simulations of D_2_R monomer, with bound spiperone or clozapine, and D_2_R homodimer without bound antagonist were performed using VMD software v1.9.2 [92] (Chicago, IL, USA). In detail, root mean square deviation (RMSD) measurements of the backbone of the transmembrane domain (TMD) of D_2_R was performed to observe receptor conformational change with respect to the initial D_2_R monomeric crystal structure (PDB id: 6CM4) [55] or initial D_2_R homodimer model. Likewise, RMSD measurements of either clozapine or spiperone in their respective MD simulations were used to monitor ligand stability in the orthosteric pocket of the D_2_R monomer. Residues in close contact (protein-ligand distance <3.5 Å) with co-crystallized ligand risperidone were compared, in terms of RMSD with MD conformations of D_2_R monomer with bound stable clozapine or spiperone, to observe differences between induced-fit of both ligands. Similarly, residues frequently close-contacted by either clozapine or spiperone in respective MD simulations, within simulation time-periods where ligands remain stable, were identified with a TCL script executed in VMD [92], thus defining ligand-specific D_2_R orthosteric pockets. After visual comparisons of the D_2_R monomer, with bound spiperone or clozapine, and D_2_R homodimer conformations, we performed an analysis of Tyr199^5.48^ and Phe390^6.52^ χ1 dihedral angle conformations using an in-house custom TCL script executed in VMD [92]. An arbitrary threshold of 240° was selected to classify Tyr199^5.48^ and Phe390^6.52^ χ1 dihedral angle cis or trans-conformation (> or <240°, respectively). The proportion of each conformation was measured. Distance analyses of the interface of D_2_R homodimer were performed using the TCL script executed in VMD [92]. An energetic analysis of the D_2_R homodimer TM5–TM6–TM5–TM6 interface was performed with FoldX v.4 (Barcelona, Catalonia, Spain) [93]. Alanine scanning of D_2_R homodimer Tyr199^5.48^ and Phe390^6.52^, generating Y199A and F390A mutations, followed by energetical analysis with FoldX v.4 [93], was carried out to measure the contribution of these residues in the homodimer interface.

## Figures and Tables

**Figure 1 ijms-20-01686-f001:**
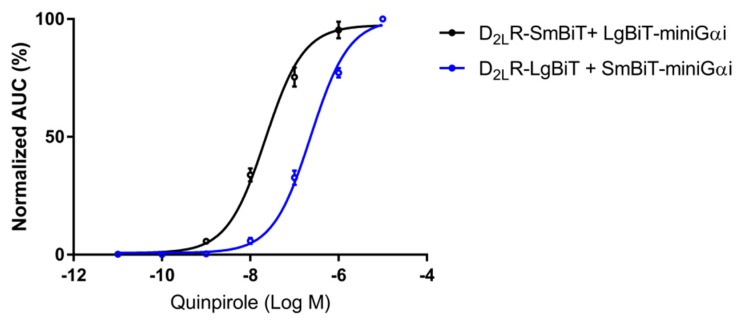
Real-time monitoring of mini-Gαi protein recruitment to the D_2L_R by the NanoLuciferase Binary Technology (NanoBiT) assay. Transient overexpression of fusion constructs of the LgBiT and SmBiT of NanoLuciferase C-terminal to D_2L_R and N-terminal to the mini-G_ai_-protein was achieved in HEK293T cells. Luminescence was monitored for 2 h. Concentration-response curves were generated by the addition of quinpirole (0.01 nM–10 µM), and the corresponding AUCs (four independent experiments, in triplicate) normalized and plotted to the logarithmic concentration of quinpirole (*n* = 12, ±SEM).

**Figure 2 ijms-20-01686-f002:**
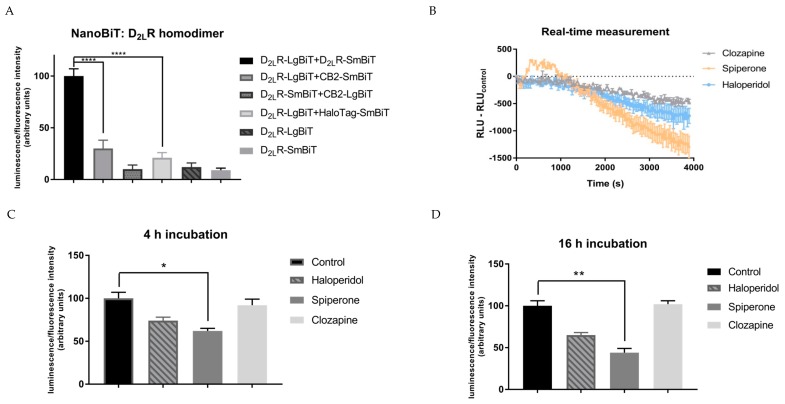
Detection of the D_2L_R homodimer using the complementation-based NanoLuciferase Binary Technology (NanoBiT). The SmBiT and LgBiT split parts of the NanoLuciferase are fused to the C-terminus of the G protein-coupled receptors (GPCRs). Overexpression of these constructs was conducted by transient transfection in HEK293T cells. The luminescent signal was normalized to the fluorescent signal of the co-transfected Venus protein in all conditions: (**A**) A non-interacting GPCR partner for D_2L_R (from a panel of multiple GPCRs) is the cannabinoid receptor 2 (CB2), which showed a 4-fold lower signal compared to the D_2L_R–D_2L_R interaction. (**B**) Real-time measurement of adherent transfected HEK293T cells incubated with 10 µM of D_2_R antagonists for 1 h. Normalized relative luminescence unit (RLU) is plotted against time (s) (*n* = 2, ±SEM). (**C**,**D**) Signals obtained following incubation of D_2L_R–LgBiT and D_2L_R–SmBiT transfected HEK293T cells with the D_2_R antagonists haloperidol, spiperone, and clozapine (10 µM) for 4 h (**C**) or 16 h (**D**). Control = solvent-treatment (DMSO ≤ 0.1%). Spiperone reduced the level of D_2L_R dimerization by ≥40% in all conditions (*n* = 5, ±SEM) (non-parametric Kruskal–Wallis one-way Anova, followed by post-hoc analysis (Dunn’s multiple comparison test), * *p* < 0.05, ** *p* < 0.01, **** *p* < 0.0001).

**Figure 3 ijms-20-01686-f003:**
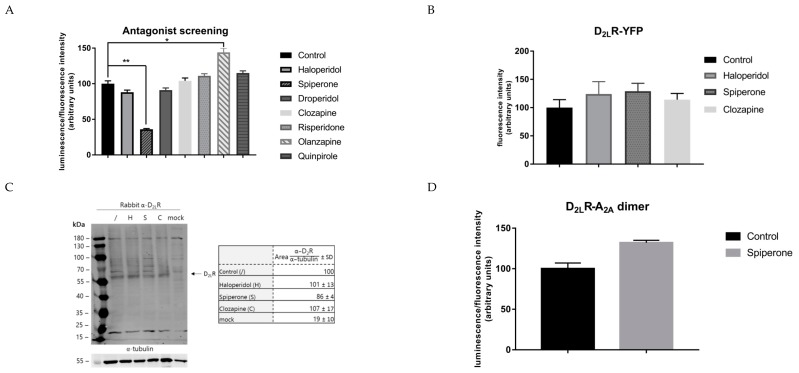
Analysis of the specificity of the effect provoked by spiperone on the D_2L_R homodimer. (**A**) A panel of different D_2_R ligands was screened via the NanoBiT assay. Only spiperone reduced the level of dimerization of D_2L_R significantly. Control = solvent-treatment (DMSO ≤ 0.1%). (**B**,**C**) To ensure the effect evoked by spiperone is not simply due to an impact on the expression levels of the D_2L_R, cells expressing D_2L_R fused to yellow fluorescent protein (YFP) were incubated with the antagonists for 16 h. No impact was observed. In addition, western blot analysis after 16 h of incubation with D_2_R antagonists also did not reveal major impact on D_2L_R expression levels (*n* = 3, ±SD) (Control = solvent-treated, H = Haloperidol, S = Spiperone, C= Clozapine, mock = non-transfected HEK293T cells). Results were normalized to tubulin values through analysis with ImageJ. Values of solvent-treated D_2L_R transfected cells were arbitrarily set as 100%. (**D**) Cells expressing another well-known dimer, A_2a_–D_2_R, were incubated with 10 µM spiperone for 16 h. Spiperone did not affect the level of A_2a_–D_2_R dimer formation. (*n* = 3, ±SEM) (non-parametric Kruskal–Wallis one-way Anova, followed by post-hoc analysis (Dunn’s multiple comparison test), * *p* < 0.05, ** *p* < 0.01).

**Figure 4 ijms-20-01686-f004:**
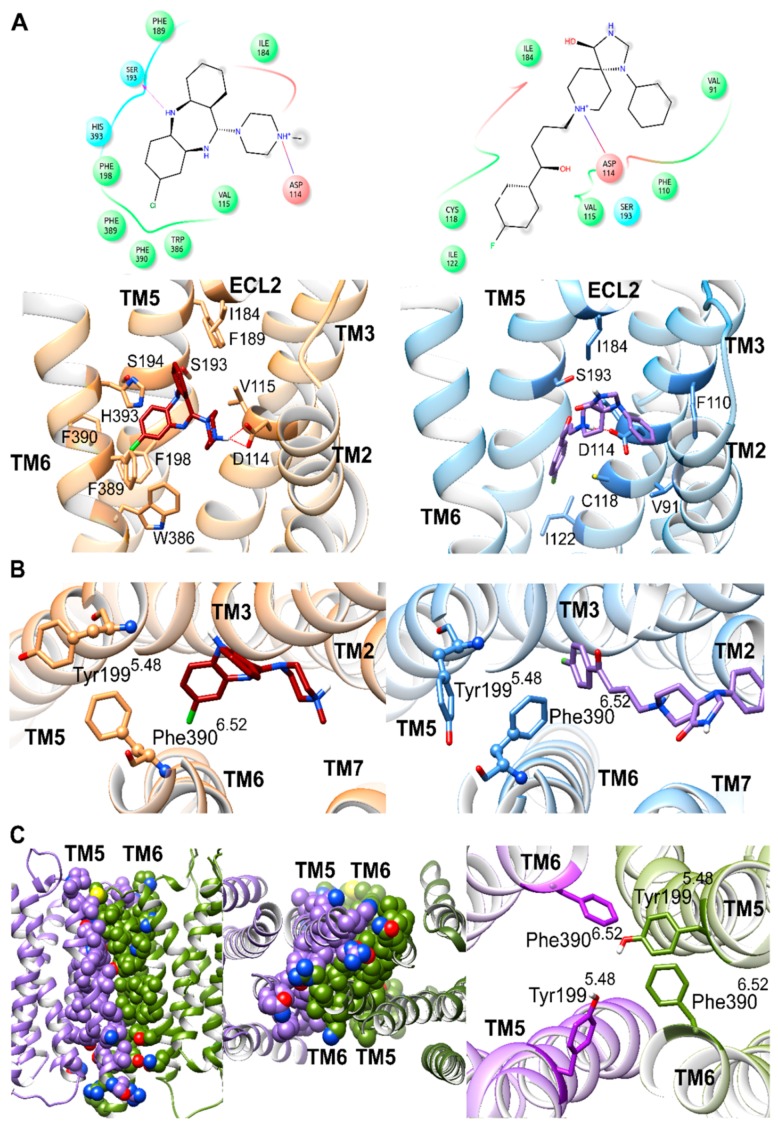
D_2_R monomer and homodimer complexes. Transmembrane helices are labeled as TM. (**A**) 2D and 3D (top and bottom, respectively) stable binding poses of residues (abbreviated following three letter or single letter code, respectively) in close contact during molecular dynamics (MD) simulation (<3.5 Å) with clozapine and spiperone (dark red and purple, respectively) bound to respective D_2_R monomer (left and right, colored in orange and blue, respectively). (**B**) Trans and cis conformations of Tyr199^5.48^ and Phe390^6.52^ χ1 dihedral angles selected by bound clozapine or spiperone (dark red and purple respectively) bound to D_2_R monomer (left and right, colored in orange and blue, respectively). (**C**) From left to right, lateral and intracellular views of TM5–TM6–TM5–TM6 D_2_R homodimer model interface, which generates aromatic interactions between Tyr199^5.48^ and Phe390^6.52^ of both D_2_R protomers during MD simulation (colored in purple or green, respectively).

**Figure 5 ijms-20-01686-f005:**
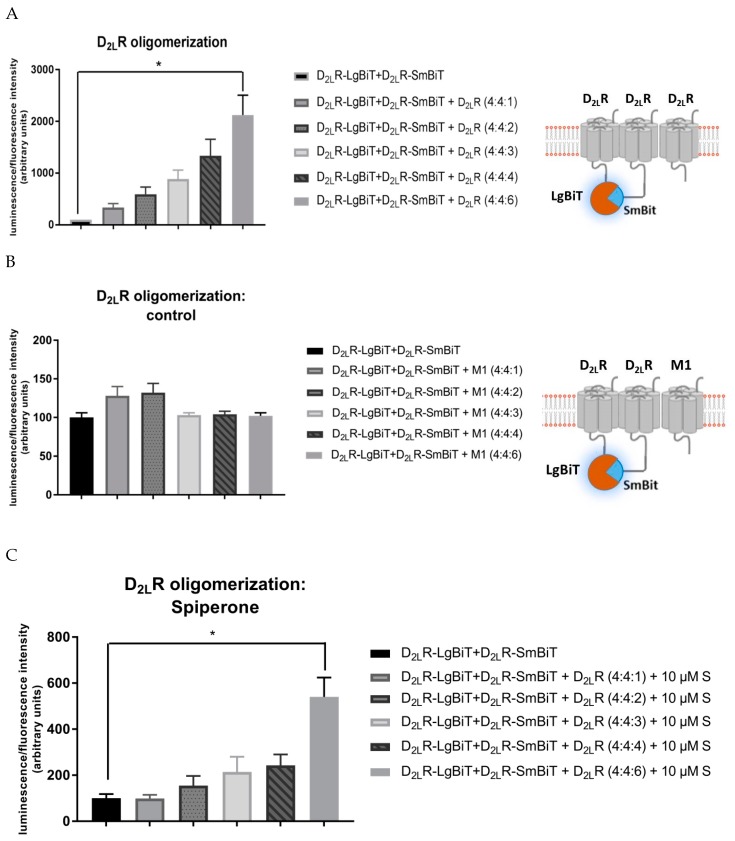
HEK293T cells transiently expressing higher levels of D_2L_R (fusion) proteins. (**A**) An increasing amount of native D_2L_R was co-transfected with D_2L_R–LgBiT and D_2L_R–SmBiT vectors. Higher expression levels of D_2L_R stimulate D_2L_R oligomerization. (**B**) An increasing amount of native muscarinic M1 receptor does not attenuate nor evoke higher luminescent signals evoked by D_2L_R dimerization. (**C**) Incubation of cells with 10 µM spiperone (S) results in a less pronounced increase of D_2L_R di- and oligomerization upon expression of increased levels of native D_2L_R. Experiments were performed three times in triplicate. (*n* = 3, ±SEM) (non-parametric Kruskal–Wallis one-way Anova, followed by post-hoc analysis (Dunn’s multiple comparison test), * *p* < 0.05).

**Table 1 ijms-20-01686-t001:** Protein–ligand interactions (<3.5 Å) of co-crystallized risperidone, and stably bound clozapine and spiperone during MD simulations. (i) Common residues in contact between all ligands; (ii) common residues in contact between risperidone and clozapine; (iii) common residues in contact between risperidone and spiperone; (iv) common residues in contact between clozapine and spiperone.

Ligand	Unique Interactions	Common Interactions	
Risperidone	Trp100^ECL1^	Asp114^3.32^	(I)
	Ser197^5.48^	Cys118^3.36^	(III)
	Phe382^6.44^	Ile122^3.40^	(III)
	Tyr416^7.43^	Trp386^6.48^	(II)
		Phe389^6.51^	(II)
Clozapine	Phe189^5.38^	Asp114^3.32^	(I)
	Ser193^5.42^	Val115^3.33^	(IV)
	Phe198^5.47^	Ile184^ECL2^	(IV)
	Phe390^6.52^	Ser193^5.42^	(IV)
	His393^6.55^	Trp386^6.48^	(II)
		Phe389^6.51^	(II)
Spiperone	Val91^2.61^	Asp114^3.32^	(I)
	Phe110^3.28^	Val115^3.33^	(IV)
		Cys118^3.36^	(III)
		Ile122^3.40^	(III)
		Ile184^ECL2^	(IV)
		Ser193^5.42^	(IV)

**Table 2 ijms-20-01686-t002:** Primers for the development of the GPCR–NanoBiT fusion constructs. **a:** Forward (F) and Reverse (R) primers (5′ > 3′) with restriction enzyme sites (**bold**), start codon (underlined) or extra nucleotides (marked in grey) to ensure a correct reading frame. **b:** Annealing temperature. **c:** Restriction enzyme.

Fusion Protein		Primers (5′ > 3′) ^a^	Tm (°C) ^b^	RE ^c^
D_2L_R-LgBiT	F	GTT**AAGCTT**ATG*AAGACGATCATC*	64	*Hin*dIII
	R	GCA**GAATTC**GC*GCAGTGGAGGATC*		*Eco*RI
D_2L_R-SmBiT	F	GTT**AAGCTT**ATG*AAGACGATCATC*	64	*Hin*dIII
	R	GCA**GAATTC**GC*GCAGTGGAGGATC*		*Eco*RI
A_2a_-LgBiT	F	CGTT**AAGCTT**ATG*AAGACGATCATCGCCCTG*	69	*Hin*dIII
	R	TGCA**GAATTC**GC*AGAAACCCCAGCACC*		*Eco*RI

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
