# Peer review of "Distinct Dopamine D2 Receptor Antagonists Differentially Impact D2 Receptor Oligomerization"

_ijms, 2019, doi:10.3390/ijms20071686_

Round 1
Reviewer 1 Report
The manuscript submitted by Elise Wouters and co-authors is a study pointing out the regulation of D2 receptor oligomerization state by D2R antagonists. For this purpose the authors developed an assay using a complementation-based Nanoluciferase Binary technology (NanoBiT). After validation of the pharmacological properties of their D2LR fusion proteins, they observed the appearance of a specific luminescence signal induced by the reassemble of the two nanoluciferase subunits when the D2LR-LgBiT and D2LR-SmBit were co-expressed in HEK-293 cells. They showed that some of D2R antagonists are able to decrease this luminescence signal. In particular they showed that the effect of spiperone could be detected after short incubation time (30 minutes) and was sustained for up 16h of incubation. The specificity of this effect has also been evaluated. Altogether, authors proposed and claimed that the decrease of luminescence signal induced by spiperone on D2R reflect a decrease in the level of D2R dimerization. Finally using molecular modelling, they proposed that the residues Tyr1995.48 and Phe 3906.52 would be involved in the homodimerization process.
Experiments are quite well performed, however, WB experiments seem to be replicated. Some conclusions are over-interpreted and not consistent with the experimental results that have been gained.
I would then recommend these minor and major changes to enhance the clarity of the manuscript and its impact:
Minor points:
1) Spiperone was used at 10µM in the experiments. It is known to have a high affinity for D2R. A dose response curve of spiperone on dimers formation could strengthen the results and the specificity of action of spiperone.
2) It was indicated in the introduction that “agonist addition has shown to prolong the D2R dimer lifetime and increase the level of dimer formation”, what is the effect of D2 agonists on NanoBiT system?
3) What is the level of luminescence obtained for specific interactions compared to the background signal.
4) Figure 2C, the legends of the samples on the western blot should be corrected.
5) Figure 5, to better clarify the results, graph A and C could be combined.
Major points:
1) It is now established that ligands previously classified as antagonists have the ability to inhibit agonist-independent activation of receptors. They, therefore, act as inverse agonists in systems that are sufficiently sensitive to allow detection. Spiperone is one of these ligands. Authors should introduce this point. In addition, it would be interesting to test the level of the constitutive activity of D2R in their system and discuss their results in view of this data.
2) On WB analysis, where are D2R monomers, dimers, or higher oligomers? Did spiperone modify the ratio monomers/dimers? How many samples have been analysed? The intensity of the bands (monomers/dimers) should be quantified to perform statistical analysis. If spiperone did not decrease dimers using WB analysis, this result is in discrepancy with results obtained with NanoBiT system. This point should be clarified and discussed.
3) Authors did not discuss the possibility that spiperone-induced changes in luminescence signal may be related to conformational changes within the dimers, inducing an increase distance between the two partners rather than from an equilibrium between monomers and dimers. This should be considered carefully.
4) Using computational techniques, the authors proposed that some residues could contribute to the dimerization interface. It would be interesting to check the model by testing the effect of mutations on the residues involved in the interface using the NanoBiT system.
Author Response
We wish to thank the reviewer for the detailed reading, the constructive and insightful comments and for the overall positive assessment of our manuscript. By complying to all reviewers’ comments we feel that the quality of our manuscript has further improved. Below we have addressed point-by-point all the comments that were raised.
The response to the reviewer is uploaded as a PDF file.

Reviewer 2 Report
Dopamine D2-like GPCRs have been associated with several central nervous system diseases. An increase in the formation of D2 homodimers has recently been associated with the pathophysiology of schizophrenia. Thus, these receptors are crucial drug targets. Understanding and targeting the modulation of D2R dimerization might help better understand the signaling behavior of dimers and their role in physiological conditions and pathophysiology of diseases. In this article, the authors used complementation-based NanoLuciferase Binary Technology assay and screened a panel of 6 D2R antagonists to understand their ability to modulate the level of D2LR dimer formation. Authors observed D2R antagonist spiperone decreased the level of D2LR dimer formation significantly by more than 40% in short and long-term incubations, while other antagonists either had minimal or no effects or even opposite effects.
The study is well planned and thorough. Below are my comments/suggestions-
Comment 1: Introduction can be edited to shorten.
Comment2: Line 55-56 – “It was first reported in 1996………Co-IP.” Need to change sentence structure, else it conveys the wrong information. D2Rs don’t form dimers by Co-IP.
Comment3: Figure 1-authors should use different symbols for different conditions.
Comment 4: Line 167 – “Again, a response…background” – Change sentence structure.
Comment 5: for the bar graphs, patterns can be used to help distinguish between different shades of grayscale.
Comment 6: Figure 2B – Include a baseline to show the decay of the NanoGlo substrate without any antagonist.
Comment 7: Line 203, Section 2.3.3 – incubation time is not included.
Comment 8: 230-231 – “ Western blot……software” – Move to the materials section.
Comment 9: Line 360 – Change HEK29T to HEK293T
Comment 10: Does spiperone effect monomer movement?
Author Response
We wish to thank the reviewer for the detailed reading, the constructive and insightful comments and for the overall positive assessment of our manuscript. By complying to all reviewers’ comments we feel that the quality of our manuscript has further improved. Below we have addressed point-by-point all the comments that were raised.
The response to the reviewer is uploaded as a PDF file

Round 2
Reviewer 1 Report
Authors considered carefully reviewer's comments. The reviewer thank the authors for their detailed answers. Therfore, the manuscript has been strengthened by the changes that had been done. The manuscipt should be now considered for publication.